# Identification of the KNOX Gene Family in *Salvia miltiorrhiza* Revealing Its Response Characteristics to Salt Stress

**DOI:** 10.3390/plants14030348

**Published:** 2025-01-24

**Authors:** Siqi Deng, Wenjing Ma, Chunxu Cui, Shiqian Wang, Mei Jiang

**Affiliations:** 1Key Laboratory for Natural Active Pharmaceutical Constituents Research in Universities of Shandong Province, School of Pharmaceutical Sciences, Qilu University of Technology (Shandong Academy of Sciences), Ji’nan 250014, China; 2Shandong Engineering Research Center for Innovation and Application of General Technology for Separation of Natural Products, Shandong Analysis and Test Center, Qilu University of Technology (Shandong Academy of Sciences), Ji’nan 250014, China

**Keywords:** *Salvia miltiorrhiza*, KNOX gene family, subcellular localization, expression analysis, salt stress, target genes

## Abstract

*Salvia miltiorrhiza* is a herbaceous plant that possesses significant medicinal value. Land salinization affects the growth of *S. miltiorrhiza*, resulting in a decline in its quality and yield. Knotted1-like homeobox (KNOX) genes are transcription factors that play important roles in plant growth and abiotic stress. The characteristics and functions of KNOX genes in *S. miltiorrhiza* remain unclear. Here, we identified ten KNOX genes in *S. miltiorrhiza*, all of which possess the characteristic four domains: KNOX1, KNOX2, ELK, and HD. These *SmKNOX*s were divided into two groups together with homologous genes. Cis-acting element analysis indicated all *SmKNOX*s contained elements associated with phytohormone, light, and stress response. The *SmKNOX*s show tissue-specific expression among roots, stems, leaves, and flowers. We assessed the response of the *SmKNOX*s to salt stress using quantitative RT-PCR analysis. Notably, *SmKNOX*4 expression significantly decreased within 24 h of salt exposure, while *SmKNOX*1, *SmKNOX*2, SmKNOX3, *SmKNOX*8, and *SmKNOX*9 showed significant increases. The expression of *SmKNOX*1, *SmKNOX*2, and *SmKNOX*3 was significantly positively correlated with that of their target genes, *GA20ox*1 and S11 *MYB*. These findings suggest that *SmKNOX*s and their target genes respond to salt stress, providing a foundation for studies of *SmKNOX*s and the potential genetic improvement of *S. miltiorrhiza*.

## 1. Introduction

Knotted1-like homeobox (KNOX) genes are a class of transcription factors that encode the homeodomain proteins, which belong to the three-amino-acid-loop-extension (TALE) superfamily. The KNOX gene family is a conserved gene family that contains four domains: KNOX1, KNOX2, homeodomain (HD), and ELK domain. The first KNOX gene, Knotted1 in maize, was the earliest identified in plants [1]. It was subsequently identified in many other plants, such as *Arabidopsis* [2], wheat [3], and *Moso bamboo* [4]. The KNOX gene plays an important role in plant growth and development. For example, studies on the loss-of-function mutant of the KNOX gene (*OSH*1) in rice revealed that KNOX genes play a vital role in the maintenance of the shoot apical meristem [5]. The *Arabidopsis* KNOX gene (*STM*) is involved in specifying floral meristem identity [6]. Studies on rice *HOS*59 mutants and overexpression lines have shown that the *HOS*59 gene has a negative regulatory effect on rice grain size and plant height [7].

Additionally, KNOX genes are implicated in plant responses to abiotic stresses. In *Gossypium hirsutum*, multiple KNOX genes were found to be involved in various stress responses. For example, silencing *GhKNOX*2 can enhance the salt tolerance of seedlings, while silencing *GhKNOX*10 and *GhKNOX*14 can reduce the tolerance of seedlings to salt stress [8]. Silencing *GhKNOX*4-A and *GhKNOX*22-D genes affects the growth and development of *G. hirsutum* seedlings by inducing oxidative stress in response to salt and drought stress [9]. In *Populus alba* × *P. glandulosa*, *PagKNAT*2/6b mediates changes in plant structure under drought conditions by directly targeting *PagGA20ox1* [10]. Advances in bioinformatics and molecular biology techniques have enabled extensive studies demonstrating gene expression differences in KNOX genes in drought, salt, and other adversity stresses by transcriptome or qRT-PCR experiments [11,12,13]. For example, 19 KNOX genes were identified in *D. huoshanense*, and the expression of most *DhKNOX* genes showed a significant upward or downward trend under PEG-mediated drought conditions [11]. Although the role of KNOX in plant development and stress response has been partially elucidated, the characteristics and functions of the KNOX gene family in *S. miltiorrhiza* are unclear.

*S. miltiorrhiza* is a perennial upright herbaceous plant in the Lamiaceae family, valued for its thick, fleshy roots with significant medicinal value. Since ancient times, *S. miltiorrhiza* has been used as a traditional Chinese medicine and is known for its ability to promote blood circulation and remove blood stasis [14]. Pharmacological studies have shown that the extract of *S. miltiorrhiza* root has an ameliorative effect on microcirculatory disorders and target organ damage caused by ischemia–reperfusion [15]. During the cultivation process, *S. miltiorrhiza* faces a variety of abiotic stresses, such as salt and drought, which can negatively impact its quality and yield. Several versions of the *S. miltiorrhiza* genome have been published, including line 99-3 [16], line DSS3 [17], and line shh [18]. These genome versions facilitate the identification and functional study of genes in *S. miltiorrhiza*.

In this study, we conducted a genome-wide identification of all KNOX genes in *S. miltiorrhiza*. We analyzed the gene structure, conserved motifs, and phylogenetic relationships of these KNOX genes, while also exploring their differentiation and function. Furthermore, we examined the expression profiles of KNOX genes in different tissues and under salt stresses. Our findings contribute to a deeper understanding of KNOX genes and the utilization of these genes to optimize the growth and development of *S. miltiorrhiza*, enabling it to better cope with environmental stress.

## 2. Results

### 2.1. Identification of SmKNOX Gene Family of S. miltiorrhiza

A combination of conserved structural domains and homologous sequence comparison analysis was applied for gene family identification. Redundant sequences were removed, and a total of ten KNOX genes were identified in *S. miltiorrhiza* (Appendix A). The analysis of basic physical and chemical characteristics indicated that the *S. miltiorrhiza* KNOX encoded amino acids with lengths ranging from 260 aa (*SmKNOX*5) to 383 aa (*SmKNOX*4), while they had molecular weights ranging from 30,056.95 (*SmKNOX*5) to 43,083.98 (*SmKNOX*4) (Table 1). The theoretical isoelectric points ranged from 4.91 (*SmKNOX*6) to 6.50 (*SmKNOX*5), with all proteins being acidic. The instability index was more than 40, except *SmKNOX*10, indicating poor protein stability. The average hydrophilicity was less than 0, suggesting the proteins were hydrophobic. We performed subcellular localization prediction based on amino acid sequences using WoLF PSORT, and the results showed that all ten *SmKNOX* proteins were located in the nucleus. Then, we randomly selected one of them for experimental validation. The results showed that the signal of the empty vector pCAMBIA1300-35S-eGFP was mainly distributed in the cell membrane and nucleus (Figure 1). The fluorescent protein signal of the fusion vector pCAMBIA1300-35S-SmKNOX4-eGFP coincides with that of the nuclear localization. This indicates that the *SmKNOX4* protein is located in the nucleus, consistent with the predictions.

### 2.2. SmKNOX Protein Sequence Alignment Analysis

A comprehensive multiple sequence alignment revealed that all ten SmKNOX proteins encompass the complete four domains characteristic of the KNOX family (Figure 2), namely, KNOX1, KNOX2, ELK, and HD. The KNOX1 and KNOX2 domains are located at the N-terminus of the protein, while ELK and HOX are located at the C-terminus of the protein. The HD domain, which is involved in DNA binding and homodimer formation [19], showed the highest level of conservation among the ten SmKNOX proteins, which is consistent with previous reports. In addition, the conserved domains of KNOX1, KNOX2, and ELK also exhibited high conservation. The ELK domain contains three conserved core sequences of “Glu(E)-Leu(L)-Lys(K)”, which may act as a nuclear localization peptide or participate in transcriptional repression [19,20]. In *S. miltiorrhiza*, except for *SmKNOX*6 and *SmKNOX*7, which are L-L-K, all others have the conserved core sequence of E-L-K.

### 2.3. Gene Structure and Conserved Domains Analysis

Further analysis of the gene structure and the conserved domains is shown in Figure 3. First, phylogenetic analysis based on the sequences of SmKNOX proteins was carried out, and the SmKNOX proteins were divided into four subfamilies (I, II, III, and IV) (Figure 3A). Then, the online software MEME (https://meme-suite.org/meme/tools/meme, accessed on 27 July 2024) was used to predict the motifs of SmKNOX proteins by finding ten motifs, and the results showed that each SmKNOX protein contained six or seven motifs (Figure 3B). The taxa in phylogenetic analysis had similar results in motif analysis. Some motifs were common to all members, such as motif2, motif3, and motif4, and some motifs were specific to subfamily members; for example, only II subfamily members included motif7 and motif9. Motif5, motif8, and motif10 were distributed only in subfamily IV, and motif6 was only not distributed in subfamily IV. In addition, only SmKNOX6 did not include motif1 (Figure 3B). The above results indicate that the functions of SmKNOXs between different subfamilies are similar but different. The detailed sequences of the different motifs are shown in Figure 3D.

By comparing the CDs of *SmKNOX* genes with the genome sequence of the corresponding gene, the UTR, CDs, and intron distribution of *SmKNOX* genes were analyzed in depth (Figure 3C). Members belonging to the same subfamily showed roughly similar exon/intron distribution patterns in terms of the exon length and the number of introns. *SmKNOX*1, classified as subfamily I, and *SmKNOX*4, classified as subfamily IV, had no UTR. In general, *SmKNOX* genes with close evolutionary relationships within the same subfamily showed consistency in gene structure and domain distribution, supporting the existence of a close evolutionary relationship between them.

### 2.4. Phylogenetic Analysis of SmKNOXs

To understand the evolutionary relationships of KNOX among *S. miltiorrhiza* and three other plant species, the sequences of the KNOX gene were analyzed among *S. miltiorrhiza* (10 *SmKNOX*s), *Arabidopsis thaliana* (8 *AtKNAT*s), *Oryza sativa* (13 *OsKNOX*s), and *G. hirsutum* (6 *GhKNOX*s). Based on the sequence of multiple alignments, an unrooted tree was established with the maximum likelihood method in MEGA 11 (Figure 4). All KNOX proteins were divided into two subfamilies (Groups I and II), and then, group I was further divided into IA, IB, and IC. To predict the potential function of *SmKNOX*s, species with homologous genes to *S. miltiorrhiza* could be studied. In Group IA, six *OsKNOX*s, *AtKNAT*1, and *GhKNOX14*-A were clustered with two SmKNOX genes, which were *SmKNOX*2 and *SmKNOX*3. In Class IB, *GhSTM*3-A and *AtSTM* were clustered with *SmKNOX*1. In Class IC, two *GhKNOX*s, two *OSKNOX*s, and two *AtKNAT*s were clustered with two *SmKNOX*s, which were *SmKNOX*6 and *SmKNOX*7. In Class II, five *OSKNOX*s, four *AtKNAT*s, and two *GhKNOX*s were clustered with *SmKNOX*4, *SmKNOX*8, and *SmKNOX*9. *GhSTM*3-A had higher homology to *SmKNOX*1 (Figure 4). *GhSTM*3 affects the floral transition of cotton [8]. Meanwhile, *SmKNOX*1 had the highest relative expression level in flowers, suggesting *SmKNOX*1 may regulate flowering time and flower development.

### 2.5. Cis-Acting Element Analysis

Cis-acting elements play a crucial role in the transcriptional regulation of gene expression, acting as key factors in enabling genes to adapt to environmental changes and regulate growth and development. In this study, we analyzed the potential cis-acting elements in the 2000 bp sequence upstream of the *SmKNOX* genes, identifying a total of 264 elements (Figure 5). These cis-acting elements were categorized into four functional groups: phytohormone response, stress response, growth and development, and light response. Notably, no growth and development-related elements were predicted for *SmKNOX*2, *SmKNOX*6, and *SmKNOX*10, while all other genes contained elements from all four categories. The distribution of these elements varied, with 136 related to light response, 80 to hormones, 34 to stress, and 14 to growth and development. Among these, light-response- and phytohormone-related elements constituted the majority, underscoring the significant roles that light and plant hormones play in the regulation of *SmKNOX* genes.

### 2.6. The Expression Profiling of SmKNOXs Genes Among Tissues

To investigate the diverse functions of *SmKNOX* genes during the development of various tissues, we analyzed the gene expression profiles among root, stem, leaf, and flower by qPCR. Previous studies have indicated that KNOX genes belonging to group I are crucial in the formation of the shoot apical meristem and the development of inflorescence structures [6,21]. Consistently, in *S. miltiorrhiza*, genes within group I (*SmKNOX*1, *SmKNOX*2, *SmKNOX*3, *SmKNOX*5, *SmKNOX*6, *SmKNOX*7, and *SmKNOX*10) were almost not expressed in the leaf. Conversely, genes categorized under subgroup II (*SmKNOX*4, *SmKNOX*8, *SmKNOX*9) were expressed in all four tissues (Figure 6A–J). The cluster heat map revealed distinct expression patterns of *SmKNOX* genes in different tissues (Figure 6K). According to the expression profiles, *SmKNOX*2, *SmKNOX*3, *SmKNOX*6, and *SmKNOX*7 were grouped into a category where these genes were predominantly expressed in the stem. *SmKNOX*1 and *SmKNOX*10 were grouped into a category where these genes were predominantly expressed in the flower. *SmKNOX*5 and *SmKNOX*8 were clustered together with the highest expression in the root, whereas *SmKNOX*4 and *SmKNOX*9 were clustered together, showing elevated expression in the leaf. This study provides valuable insights for understanding the functional roles of these genes in tissue development.

### 2.7. Expression Analysis of SmKNOXs Under Salt Stress

To elucidate the significant role that *KNOX* genes in *S. miltiorrhiza* may play in responding to salt stress, we exposed the seedlings to 100 mM NaCl and monitored the changes in gene expression at intervals of 0 h, 3 h, 6 h, 24 h, and 48 h (Figure 7). Upon salt treatment, the expression patterns of *SmKNOX*1, *SmKNOX*2, and *SmKNOX*3 exhibited a similar up–down–up trend, peaking at 24 h with a significant increase compared to the 0 h baseline. The expression level of the *SmKNOX*8 gene consistently and significantly rose following salt exposure, reaching its maximum at 48 h. In contrast, the *SmKNOX*9 gene showed a substantial increase at 3 h post-treatment, after which, it gradually declined. On the other hand, *SmKNOX*4 expression significantly decreased after salt stress but recovered by the 48 h. The cluster heat map revealed that *SmKNOX*4, *SmKNOX*7, and *SmKNOX*8 formed a group with higher expression levels observed at 48 h. Another group, comprising *SmKNOX*1, *SmKNOX*2, *SmKNOX*3, and *SmKNOX*5, showed higher expression at 24 h. Lastly, the group including *SmKNOX*6, *SmKNOX*9, and *SmKNOX*10 had higher expression levels at 3 h. Overall, under salt stress, the expression levels of *SmKNOX* genes varied, with a significant decrease in *SmKNOX*4 and a significant increase in *SmKNOX*1, *SmKNOX*2, *SmKNOX*3, *SmKNOX*8, and *SmKNOX*9.

### 2.8. Expression Analysis of Target Genes of SmKNOXs Under Salt Stress

KNOX is a transcription factor that regulates plant growth and responses to abiotic stress by controlling the expression of downstream target genes [22]. According to previous studies, KNOX target genes include *REVOLUTA*, *ABI*3, *GA20ox*1, and S11 *MYB* [23,24,25,26]. We analyzed the 2000 bp upstream sequences of these target gene coding sequences (CDs) extracted from the *S. miltiorrhiza* genome. The results showed that all sequences contained the TGAC core binding motif of the KNOX protein (Appendix A). The qPCR analysis revealed that, except S11 *MYB*-2, the expression levels of the other target genes were significantly upregulated under salt stress (Figure 8). Among these, *ABI*3, *GA20ox*1, S11 *MYB*-1, S11 *MYB*-2, and S11 *MYB*-3 reached their highest expression levels at 24 h and then decreased, while *REVOLUTA*-1 and *REVOLUTA*-2 showed peak expression levels at 48 h. Correlation analysis of *SmKNOX* genes and their target genes demonstrated that the expression of *SmKNOX*1 was significantly positively correlated with the expression of S11 *MYB*-2 (r ≥ 0.9, *p* ≤ 0.05). Similarly, the expression levels of *SmKNOX*2 and *SmKNOX*3 were significantly positively correlated with the expression of *GA20ox*1 and two S11 MYB genes (r ≥ 0.9, *p* ≤ 0.05). *SmKNOX*5 exhibited a significant positive correlation with the expression of *ABI*3, *GA20ox*1, and two S11 *MYB* genes (r ≥ 0.9, *p* ≤ 0.05). Additionally, *SmKNOX1*, *SmKNOX2*, and *SmKNOX3* were significantly upregulated under salt stress. These results suggest that KNOX likely responds to salt stress and positively regulates the expression of downstream target genes, including *GA20ox*1 and S11 *MYB*.

### 2.9. Gene Cloning of SmKNOXs

To obtain the CDs of *SmKNOX* genes in *S. miltiorrhiza*, gene cloning was performed. By aligning these sequences with the reference genome, we found that the *SmKNOX* genes displayed single-nucleotide polymorphisms (SNPs) and insertions and deletions (indels), indicating individual variability among *S. miltiorrhiza* plants (Appendix A). Based on alignment results, we categorized the 10 *SmKNOX* genes into three groups. The first group, including *SmKNOX*1, *SmKNOX*2, *SmKNOX*4, *SmKNOX*8, and *SmKNOX*10, had synonymous SNPs that did not alter the amino acid sequence. The second group, comprising *SmKNOX*5 and *SmKNOX*7, had non-synonymous SNPs that changed the amino acid sequence. The third group, which included *SmKNOX*3, *SmKNOX*6, and *SmKNOX*9, exhibited indels (Figure 9). Specifically, *SmKNOX*3 had a three-base deletion between 144 bp and 146 bp; *SmKNOX*6 had two indels with three bases inserted between 134 bp and 135 bp and twelve bases inserted between 366 bp and 377 bp; *SmKNOX*9 had a deletion of 114 bases between 157 bp and 270 bp. Despite these indels, they only affected the length of the amino acid sequence without impacting the subsequent encoded amino acids. This variability could be due to individual differences in *S. miltiorrhiza* plants or potential errors in genome annotation. The cloning results of *SmKNOX* genes in *S. miltiorrhiza* provide a reference sequence for further studies on these genes.

## 3. Discussion

The homeobox gene KNOX transcription factor family is a subset of the TALE supergene family, playing a crucial role in various biological processes such as plant growth and development [22,27]. To date, members of the KNOX gene family have been identified in multiple plants, including *Arabidopsis*, rice, and cotton, with their functions well characterized [2,6,7,8]. However, no KNOX genes have been reported in the medicinal plant *S. miltiorrhiza*. Additionally, previous research has mainly focused on the impact of KNOX genes on plant development, with limited studies examining their role in responding to abiotic stress. To better understand the potential functions of KNOX genes in *S. miltiorrhiza*, this study employed bioinformatics and molecular biology techniques to investigate *SmKNOX* genes. We examined the gene structure, phylogeny, expression patterns, and response to salt stress. These findings offer insights into the functions of KNOX genes in *S. miltiorrhiza*.

KNOX genes are typically classified into two subfamilies, class I and class II, based on their structural characteristics, expression patterns, and phylogenetic relationships [28,29]. In this study, we constructed a phylogenetic tree containing 10 *SmKNOX* genes from *S. miltiorrhiza* and 37 *KNOX* genes from various species. The results indicated that *SmKNOX1*, *SmKNOX2*, *SmKNOX3*, *SmKNOX5*, *SmKNOX6*, *SmKNOX7*, and *SmKNOX10* in *S. miltiorrhiza* belong to class I. Different KNOX genes have distinct biological functions. Class I KNOX proteins are essential for the formation of shoot apical meristems, internode elongation, and inflorescence structure in angiosperms [6,21,30]. In *S. miltiorrhiza*, class I KNOX genes were highly expressed in flowers or stems but rarely in leaves, similar to the expression pattern observed in *Arabidopsis* and maize [27], where class I genes are mainly expressed in the shoot apex, stem, and inflorescence shoot apex. This suggests that class I KNOX proteins in *S. miltiorrhiza* may play a role in shoot apex and inflorescence development. In the phylogenetic tree, *SmKNOX*6, 7, and 10 of *S. miltiorrhiza* clustered with *AtKNAT*2, *AtKNAT*6, and *GhKNOX*2-A. Previous studies have shown that *AtKNAT*2 and *AtKNAT*6 negatively regulate the development of inflorescence structure [31,32]. However, in *S. miltiorrhiza*, *SmKNOX*6, and *SmKNOX*7 were expressed at higher levels in stems, and only *SmKNOX*10 was highly expressed in flowers. *SmKNOX*4, *SmKNOX*8, and *SmKNOX*9 in *S. miltiorrhiza* belong to class II. Class II KNOX genes play a significant role in vascular tissue development and lateral organ differentiation [7,33]. Based on the expression pattern, class II *SmKNOX* genes of *S. miltiorrhiza* show high expression levels in roots, stems, leaves, and flowers, which aligns with observations in rice, maize, and poplar [27]. Generally, class II KNOX genes are more broadly expressed in various tissues than class I KNOX genes. Phylogenetic analysis revealed that *SmKNOX*4 of *S. miltiorrhiza* clustered with *AtKNAT3* and *AtKNAT4*. Previous studies have demonstrated that *AtKNAT*3 synergistically influences the biosynthesis of secondary cell walls, thereby altering the mechanical support strength of *Arabidopsis* stems [33,34]. This study provides a direction for further research on the function of KNOX genes in *S. miltiorrhiza* through phylogenetic relationship and expression pattern analysis, with future molecular biology experiments needed to verify their functions.

Abiotic stress, such as salt stress, can significantly reduce productivity and lower plant yields [35]. KNOX genes play a crucial role in responding to abiotic stress [36,37]. In this study, we assessed the response of the *SmKNOX* gene to salt stress using quantitative RT-PCR analysis. Notably, the expression of the *SmKNOX*4 gene decreased significantly within 24 h of salt stress, which may help to enhance the resistance of *S. miltiorrhiza* to salt stress. Conversely, other genes showed increased expression within 24 h, with *SmKNOX*1, *SmKNOX*2, *SmKNOX*8, and *SmKNOX*9 exhibiting significant upregulation, suggesting their broad mobilization in response to salt stress in *S. miltiorrhiza*. In *G. hirsutum*, silencing the *GhKNOX*14 gene decreases seed tolerance to salt [8]. Phylogenetic tree analysis revealed that *SmKNOX*2 is closely related to *GhKNOX*14. The expression of *SmKNOX*1 and 2 did not change notably at 3 h and 6 h of salt stress but significantly increased at 12 h before decreasing after 48 h. In *G. hirsutum*, silencing the *GhKNOX*4-A gene significantly impacts seedling growth under salt treatment [9]. Phylogenetic tree analysis revealed that *SmKNOX*8 and *SmKNOX* 9 are closely related to *GhKNOX*4-A. This indicates that these genes may have important roles in the salt stress response of *S. miltiorrhiza*. However, further systematic studies are needed to determine if their roles are identical to those of *GhKNOX*14 in *G. hirsutum*. In conclusion, the *SmKNOX* gene has a potential regulatory function in responding to salt stress, providing a foundation for future functional research on the *SmKNOX* gene and promoting the genetic improvement of *S. miltiorrhiza* under salt stress.

## 4. Materials and Methods

### 4.1. Plant Materials

*S. miltiorrhiza* plants were cultivated in the laboratory of Qilu University of Technology, Changqing District, Jinan City, Shandong Province, China (geospatial coordinates: E 116.8178, N 36.5645), and identified by Liu Wei, a researcher at Qilu University of Technology. Fifteen uniformly growing seedlings were selected for the salt stress experiment.

### 4.2. Gene Family Identification and Characterization

To identify the KNOX gene family members in *S. miltiorrhiza*, the *S. miltiorrhiza* genome sequence was obtained from the National Genomics Data Center with the accession number of GWHDOEA00000000 and described in a previously published paper [16]. We identified the KNOX gene family members based on the conserved structural domains and sequence homology. First, we downloaded the Hidden Markov Model (HMM) corresponding to four conserved structural domains from the Pfam database [38], which were the ELK domain (PF03789), KNOX1 domain (PF03790), KNOX2 domain (PF03791), and Homeobox KN domain (PF05920). The protein sequences of all genes within the *S. miltiorrhiza* genome were searched against these four HMMs separately by hmmsearch software (v. 3.3.2) under default parameters. Genes with four conserved structural domains at the same time were used as candidate KNOX gene family members. Then, we downloaded the protein sequences of *A. thaliana*’s KNOX genes from the UniprotKB database [39]. The protein sequences of all genes within the *S. miltiorrhiza*’s genome were against *A. thaliana*’s KNOX sequences by BLASTP software (v. 2.14.0) [40] at the parameter e-value ≤ 1 × 10^5^. Finally, genes that met both the domains and the sequence homology were identified as members of the *S. miltiorrhiz* KNOX gene family.

The basic physicochemical properties of the protein sequences, including theoretical pI, instability index (II), and grand average of hydropathicity (GRAVY), were analyzed by ProtParam on the ExPASy website (https://www.expasy.org/, accessed on 27 July 2024). The structural features of the genes were analyzed and visualized by TBtools software (v. 2.136) [41] based on the annotation files of *S. miltiorrhiza*’s genome. Conserved structural domains of protein sequences were analyzed through the Multiple Expectation Maximization for motif Elicitation (MEME) website (https://meme-suite.org/meme/tools/meme, accessed on 27 July 2024) [42] and visualized by TBtools software (v. 2.136). To analyze the cis-acting elements, we obtained sequences located 2 kb upstream of the start codon (ATG) of the *SmKNOX* gene, which was subsequently analyzed using the PlantCARE tool (https://bioinformatics.psb.ugent.be/webtools/plantcare/html/, accessed on 28 July 2024). Finally, we visualized the results using TBtools (v. 2.136).

### 4.3. Multiple Sequence Alignment and Phylogenetic Relationship Analysis

The sequences of the SmKNOX proteins of *S. miltiorrhiza* were subjected to sequence alignment by the ClustalW tool of MEGA software (v. 11) [43] under default parameters. The alignment results were visualized by GeneDoc software (v. 2.7). To analyze the phylogenetic relationships among SmKNOX proteins, the sequence alignment results were further used for phylogenetic tree construction using MEGA software (v. 11) [43]. The phylogenetic tree construction was performed using the maximum likelihood (ML) method with bootstrap replications set to 1000.

To analyze the phylogenetic relationships of KNOX genes in *S. miltiorrhiza*, *A. thaliana*, *O. sativa,* and *G. hirsutum*, we downloaded the protein sequences of KNOX genes belonging to *A. thaliana*, *O. sativa*, and *G. hirsutum* from Phytozome v13 website (https://phytozome-next.jgi.doe.gov/, accessed on 27 July 2024) [44], respectively. The protein sequences were subjected to alignment by the ClustalW tool of MEGA software (v. 11) under default parameters, and then, the sequence alignment results were further used for phylogenetic tree construction using MEGA software (v. 11). The phylogenetic tree construction was performed using the ML method with bootstrap replications set to 1000. Finally, the phylogenetic tree was visualized through the ChiPlo website (https://www.chiplot.online/, accessed on 28 July 2024) [45].

### 4.4. Tissue Differential Expression Analysis of SmKNOX Genes

*S. miltiorrhiza* plants were planted in the laboratory (School of Pharmaceutical Sciences, Qilu University of Technology) and grown under natural conditions for two years. During the flowering period, the root, stem, leaf, and flower of *S. miltiorrhiza* were collected from three separate plants. These samples were immediately frozen in liquid nitrogen and stored at −80 °C. The tissue expression profile of the *SmKNOX* gene was analyzed by qRT-PCR. The *SmActin* gene was selected as an internal reference gene according to a previous paper [46]. The primers are shown in Appendix A. The relative expression of *SmKNOX*s among different tissues was analyzed using the 2^−ΔΔCt^ method.

### 4.5. Stress Response Analysis of SmKNOX Genes

Fifteen annual seedlings with consistent growth were used for salt stress treatment. The *S. miltiorrhiza* seedlings were treated with salt stress using 100 mM NaCl. They were planted in soil and cultured in a climate chamber at 25 °C with a day-to-night ratio of 16:8. Leaf samples were collected from the *S. miltiorrhiza* plants at 0 h, 3 h, 6 h, 24 h, and 48 h during the salt stress treatment. The samples were stored immediately in a −80 °C refrigerator until use. Three biological replicates were used per treatment.

RNA was extracted using the M5 Plant RNeasy Complex Mini Kit (Mei5 Biotechnology Co., Ltd., Beijing, China). The integrity and concentration of RNA was checked using agarose electrophoresis gel and a NanoDrop 2000 spectrophotometer (Thermo Scientifc, Waltham, MA, USA), respectively. RNA was reverse transcribed into cDNA using the Hifair^®^ II 1st Strand cDNA Synthesis SuperMix for qPCR (Yeasen Biotech, Shanghai, China) with Oligo (dT) as a primer. The qRT-PCR primers were designed using the IDT website and are shown in Appendix A. The qRT-PCR experiments were performed using the Hieff qPCR SYBR Green Master Mix Kit (Yeasen Biotechnology, China) and detected by ABI QuantStudio 5 (Thermo Fisher, Waltham, MA, USA). The *SmActin* gene was selected as the internal reference gene as documented previously [46], and the relative expression of *SmKNOX*s among different treatments was analyzed by the 2^−ΔΔCt^ method.

### 4.6. Gene Cloning Analysis of SmKNOX Genes

Gene cloning analysis was performed using pCAMBIA1300-35S-eGFP vectors, which were linearized by BamHI enzymes and SalI enzymes. The gene cloning primers consisted of two parts, the vector terminal homologous sequence and the gene sequence, and are shown in Appendix A. The *SmKNOX* sequence was amplified from the cDNA by high-fidelity enzymes, and the amplified length was detected by agarose electrophoresis gel. Bands meeting the expected length were recovered using the DNA Gel Extraction Kit with Magnetic Beads (Beyotime, Shanghai, China) to obtain the purified sequence. The purified sequence was ligated to the linearized vector using a ClonExpress^®^ II One Step Cloning Kit (Vazyme, Nanjing, China). The colonies were selected for PCR amplification and Sanger sequencing verification.

### 4.7. Subcellular Localization Analysis

The fusion vector pCAMBIA1300-35S-SmKNOX-EGFP was constructed. An empty pCAMBIA1300-35S-EGFP vector was used as a control. The vector was transformed into Agrobacterium GV3101 using the freeze–thaw method. After two weeks of tobacco seed germination, seedlings of uniform size were selected for transplanting and grown for four weeks in an incubation chamber at 25 °C, 16 h light/8 h dark. Agrobacterium tumefaciens was injected into the leaf epidermis of four-week-old tobacco for transient expression. The expression of the fusion protein was observed and photographed using laser confocal microscopy.

## 5. Conclusions

Here, the KNOX gene family was identified in *S. miltiorrhiza*, and a comprehensive analysis was conducted on gene structure, conserved domains, cis-acting elements, gene cloning, and expression patterns across different tissues and under salt stress conditions. A phylogenetic tree, incorporating 10 *SmKNOX* genes from *S. miltiorrhiza* and 27 KNOX genes from various species, alongside expression pattern analysis, revealed that group I KNOX proteins in *S. miltiorrhiza* may be involved in shoot apex and inflorescence development. Under salt stress, the expression levels of *SmKNOX*1, *SmKNOX*2, *SmKNOX*3, *SmKNOX*8, and *SmKNOX*9 increased significantly, while *SmKNOX*4 exhibited a marked decrease within 24 h of exposure. Furthermore, the expression of *SmKNOX*1, *SmKNOX*2, and *SmKNOX*3 was significantly positively correlated with that of their target genes, *GA20ox*1 and S11 *MYB*. In summary, *SmKNOX1*, *SmKNOX2*, and *SmKNOX3* respond to salt stress and may further positively regulate target gene expression.

## Figures and Tables

**Figure 1 plants-14-00348-f001:**
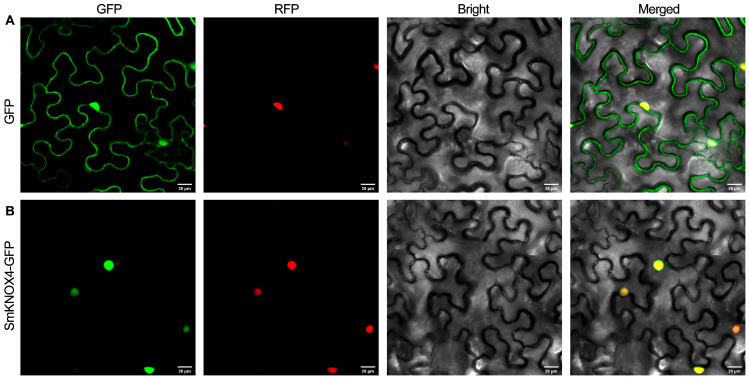
Subcellular localization results. (**A**,**B**) represent empty vectors and *SmKNOX4* proteins, respectively. From left to right, the images represent GFP fluorescence, RFP fluorescence, bright field, and merged images.

**Figure 2 plants-14-00348-f002:**
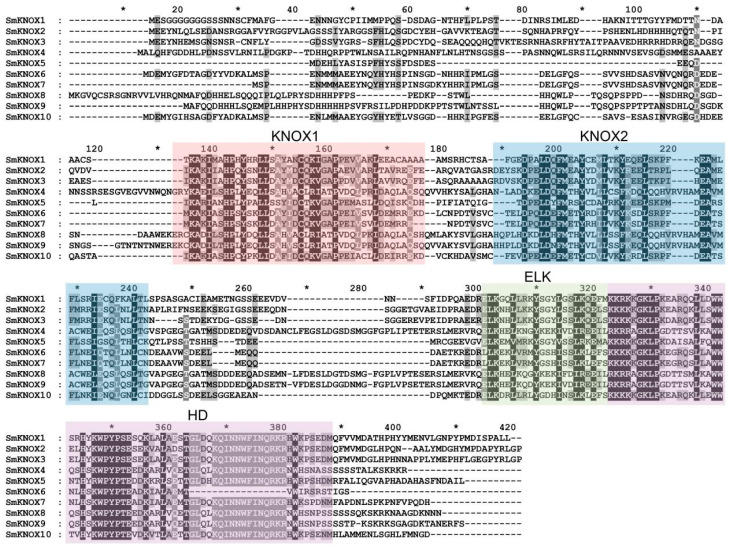
Multi-sequence alignment of conserved domains within the *SmKNOX* proteins. The KNOX1, KNOX2, ELK, and HD motifs are denoted by red, blue, green, and purple rectangles, respectively. The asterisk above the sequence represents the middle of the two numbers.

**Figure 3 plants-14-00348-f003:**
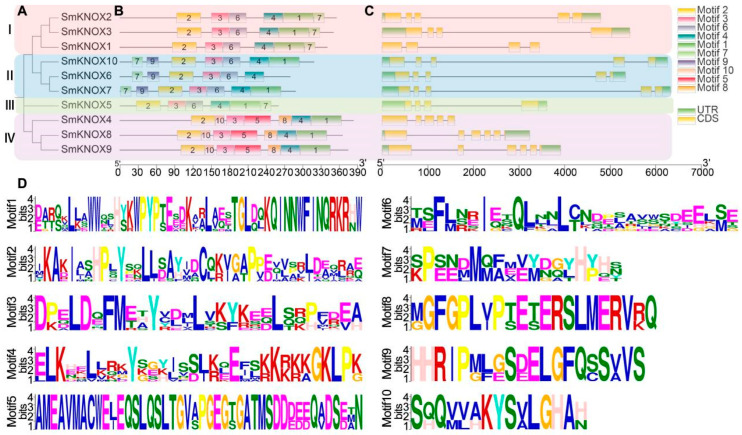
Gene structure, conserved domain, and motif analysis of SmKNOXs proteins from *S. miltiorrhiza*. (**A**) The phylogenetic tree of all SmKNOXs proteins was constructed using the ML method. (**B**) Motifs in the SmKNOX proteins were identified by the MEME program. Motifs numbered 1–10 were colored differently. (**C**) The UTR, CDs, and intron organization of *SmKNOX*s. The green boxes represent UTRs, the yellow boxes represent CDs, and thin black lines represent introns. (**D**) The abscissa of the sequence logos refers to the amino acid with the highest frequency, and the ordinate represents the relative frequency of the corresponding amino acid.

**Figure 4 plants-14-00348-f004:**
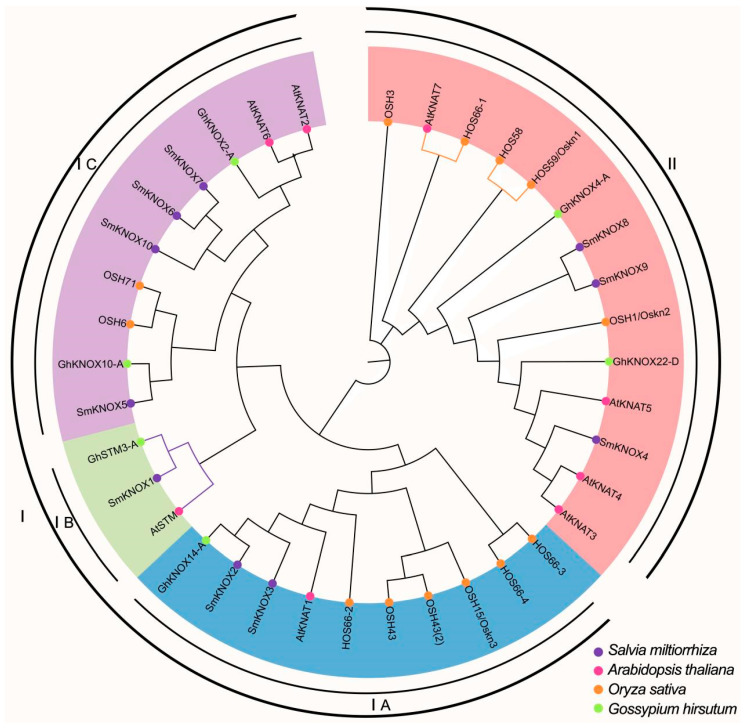
Unrooted phylogenetic tree of KNOX proteins from *S. miltionhiza*, *A. thaliana*, *O. sativa*, and *G. hirsutum*. The maximum likelihood (ML) phylogenetic tree was constructed using MEGA7.0 with 1000 bootstrap replicates. These KNOX proteins are divided into two groups (I–II) with three subgroups (IA, IB, and IC), distinguished by different colors. Genes from different species are represented by circles of different colors.

**Figure 5 plants-14-00348-f005:**
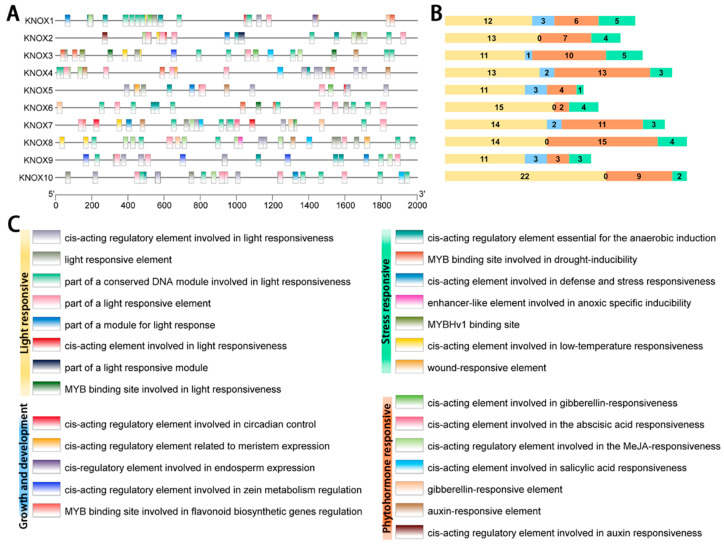
Analysis of cis-acting element in *SmKNOX* genes. (**A**) Inspection of cis-acting elements in *SmKNOX* genes. The black line indicates the promoter length of the *SmKNOX* genes. (**B**) The total number of cis-acting elements in each group. (**C**) The type of cis-acting elements belonging to each group.

**Figure 6 plants-14-00348-f006:**
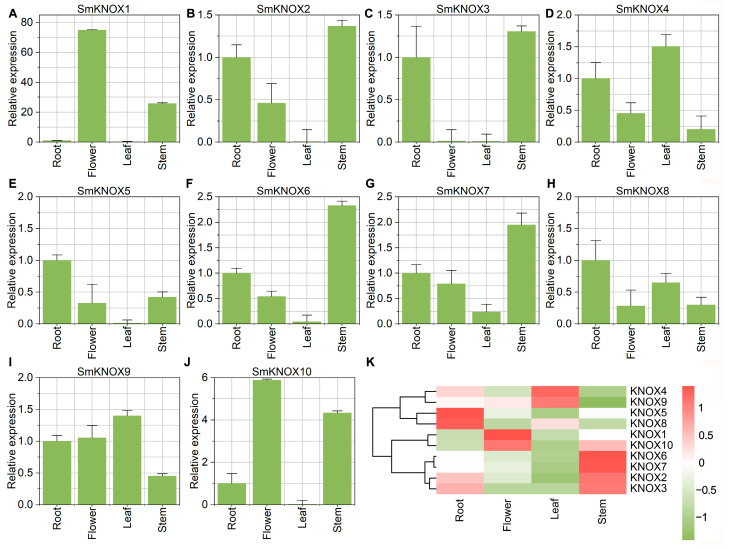
Expression profiles of *SmKNOX* genes in different tissues of *S. miltiorrhiza*. (**A**–**J**) Relative expression levels of *SmKNOX* genes in flowers, leaves, roots, and stems. The X-axis represents different tissues, and the Y-axis represents the relative expression levels normalized to the expression values of root. (**K**) A heatmap generated based on the relative expression levels of *SmKNOX* genes in different tissues. The color corresponds to the Z-score transformed from the expression levels of *SmKNOX* genes.

**Figure 7 plants-14-00348-f007:**
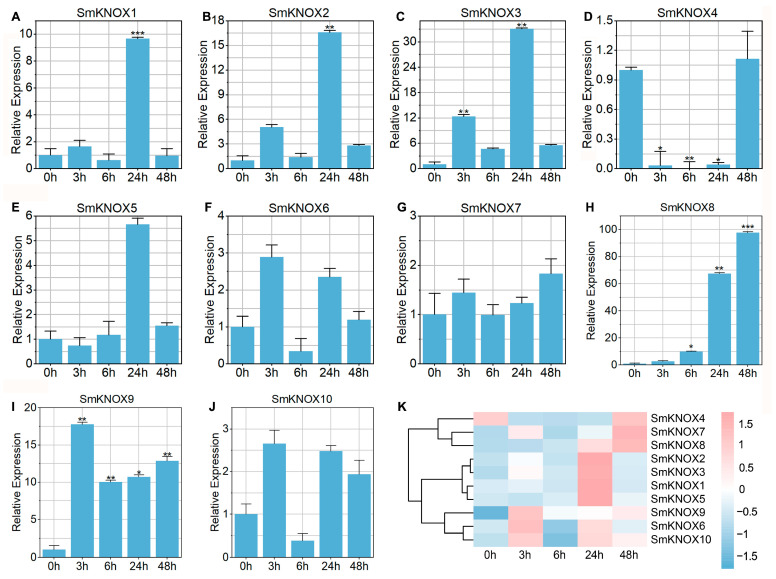
Expression profiles of *SmKNOX* genes in *S. miltiorrhiza* under different salt stress periods. (**A**–**J**) Relative expression levels of *SmKNOX* genes after 0 h, 3 h, 6 h, 24 h, and 48 h of salt stress. The X-axis represents different periods, and the Y-axis represents the relative expression levels normalized to the expression values of 0 h. Error bars represent the standard error. Asterisks represent significant changes in expression compared with 0 h; specifically, *, **, and *** represent adjusted *p* < 0.05, *p* < 0.01, and *p* < 0.001, respectively. (**K**) Heatmap generated based on the relative expression levels of *SmKNOX* genes in different periods. The color corresponds to the Z-score transformed from the expression levels of *SmKNOX* genes.

**Figure 8 plants-14-00348-f008:**
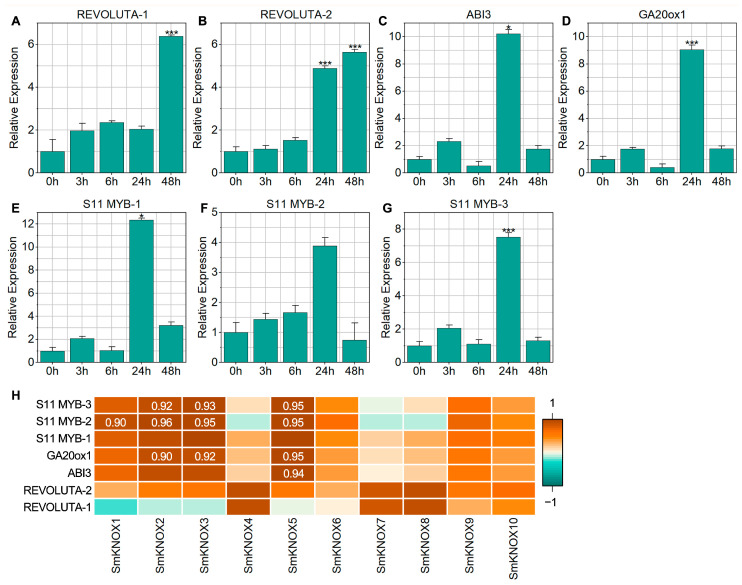
Expression profiles of target genes of *SmKNOX*s under different salt stress periods. (**A**–**G**) Relative expression levels of target genes after 0 h, 3 h, 6 h, 24 h, and 48 h of salt stress. The X-axis represents different periods, and the Y-axis represents the relative expression levels normalized to the expression values of 0 h. Error bars represent the standard error. Asterisks represent significant changes in expression compared with 0 h; specifically, * and *** represent adjusted *p* < 0.05, and *p* < 0.001, respectively. (**H**) Correlation of expression between *SmKNOX* and target genes. Numbers represent correlation values where the expression of two genes is significantly correlated (r ≥ 0.9, *p* ≤ 0.05).

**Figure 9 plants-14-00348-f009:**
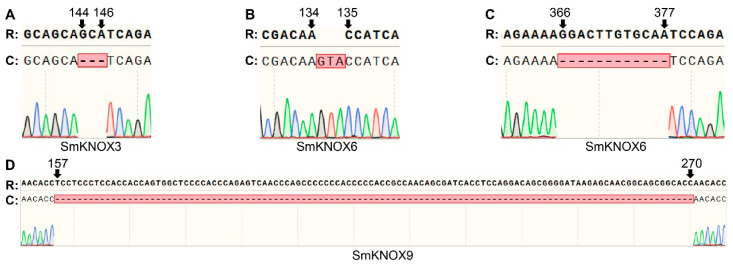
The gene cloning results for *SmKNOX*3, *SmKNOX*6, and *SmKNOX*9. (**A**) The indel fragment of the *SmKNOX*3 gene. The first (**B**) and second (**C**) indel fragments of the *SmKNOX*6 gene. (**D**) The indel fragment of the *SmKNOX*9 gene. “R” represents the CDs extracted from the reference genome, and “C” represents the CDs obtained from gene cloning.

**Table 1 plants-14-00348-t001:** Basic characteristics of the KNOX gene family in *S. miltiorrhiza*.

Gene Name	Number of Amino Acids	Molecular Weight	pI	Instability Index	Grand Average of Hydropathicity
*SmKNOX*1	340	38,093.08	5.73	57.33	−0.56
*SmKNOX*2	355	40,606.38	5.86	45.66	−0.849
*SmKNOX*3	350	40,538.12	6.22	55.54	−1.065
*SmKNOX*4	383	43,083.98	6.15	59.2	−0.72
*SmKNOX*5	260	30,056.95	6.50	52.1	−0.717
*SmKNOX*6	279	32,165.15	4.91	42.07	−0.71
*SmKNOX*7	288	33,516.71	5.50	45.65	−0.863
*SmKNOX*8	365	41,819.66	6.25	61.99	−0.924
*SmKNOX*9	374	42,534.07	6.03	53.65	−0.941
*SmKNOX*10	318	36,265.81	4.99	39.75	−0.671

## Data Availability

Data is contained within the article or Appendix A.

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
