# Peer review of "Identification of the KNOX Gene Family in Salvia miltiorrhiza Revealing Its Response Characteristics to Salt Stress"

_plants, 2025, doi:10.3390/plants14030348_

Round 1
Reviewer 1 Report (Previous Reviewer 2)
Comments and Suggestions for Authors
In this article, Deng and his (her) colleagues conducted a genome-wide identification of all KNOX genes in Salvia miltiorrhiza and examined the expression profiles of KNOX genes in different tis sues and under salt stresses to explore their function. The manuscript enriches our knowledge about the role of KNOX in S. miltiorrhiza. The conclusions are supported by the data, and the revised manuscript is improved obviously and general interest to the readers. I suggest accept the revised manuscript after minor revisions.
In scientific aspects:
1. In the section 2.2, I suggest that the “A comprehensive multiple sequence alignment of the SmKNOX proteins revealed that all 10 SmKNOX genes encompass the complete four domains……”(page4, L113-114) replaced by “A comprehensive multiple sequence alignment revealed that all the ten SmKNOX proteins encompass the complete four domains……”. Proteins consist of chains of amino acids, while genes consist of DNA or RNA. There are different, the author should check in the whole text.
Comments on the Quality of English LanguageIn language aspect:
1. I suggest that the “ The smKNOXs had……”(page1, L22) replaced by “The smKNOXs show ……”.
1. Make sure that the gene names should be in italic format (Page5, L163-164), or the “…… the KNOX gene……”(page5, L162) replaced by “…… the KNOX proteins……”.
2. There are only the examples. The authors must carefully check grammar, punctuation, spelling, and overall style in the whole text before publication.
Author Response
In this article, Deng and his (her) colleagues conducted a genome-wide identification of all KNOX genes in Salvia miltiorrhiza and examined the expression profiles of KNOX genes in different tis sues and under salt stresses to explore their function. The manuscript enriches our knowledge about the role of KNOX in S. miltiorrhiza. The conclusions are supported by the data, and the revised manuscript is improved obviously and general interest to the readers. I suggest accept the revised manuscript after minor revisions.
Thank you for your recognition of our study. We have revised the paper carefully based on your comments. We hope that our revisions have addressed all the comments satisfactorily.
Comments 1: In the section 2.2, I suggest that the “A comprehensive multiple sequence alignment of the SmKNOX proteins revealed that all 10 SmKNOX genes encompass the complete four domains……”(page4, L113-114) replaced by “A comprehensive multiple sequence alignment revealed that all the ten SmKNOX proteins encompass the complete four domains……”. Proteins consist of chains of amino acids, while genes consist of DNA or RNA. There are different, the author should check in the whole text.
Response 1: Thanks for the suggestion. We have revised this sentence in lines 105-106 of the marked manuscript and have checked and revised similar errors throughout the text.
Comments 2: I suggest that the “ The smKNOXs had……”(page1, L22) replaced by “The smKNOXs show ……”.
Response 2: Thanks for the suggestion. We have revised this sentence in line 21.
Comments 3: Make sure that the gene names should be in italic format (Page5, L163-164), or the “…… the KNOX gene……”(page5, L162) replaced by “…… the KNOX proteins……”.
Response 3: Thanks for the suggestion. We have italicized the gene names in line 156-157.
Comments 4: There are only the examples. The authors must carefully check grammar, punctuation, spelling, and overall style in the whole text before publication.
Response 4: Thanks for the suggestion. We have checked and revised the full text for grammar, punctuation, spelling and overall style.

Reviewer 2 Report (New Reviewer)
Comments and Suggestions for Authors
Dear Authors,
I am pleased to inform you that this is a well-executed manuscript. The paper is well-structured, and the topic addressed is relevant to the field.
Some remarks:
I noticed that most of the keywords are already included in the title. To enhance the discoverability of the manuscript and improve its indexing, I recommend revising the keywords to include terms or phrases that complement the title but are not explicitly mentioned in it.
Line 64 - Replace Salvia miltiorrhiza whit S. miltiorrhiza
Line 164 – Replace Gossypium hirsutum with G. hirsutum
In general, the full Latin name of each species be written out in its entirety the first time it appears in the text. Subsequent mentions can then use the abbreviated form consistently throughout the manuscript. You have largely followed this rule; however, I noticed some inconsistencies in a few instances. Please review the manuscript to ensure that this rule is applied consistently to all species, as this will enhance the uniformity and professionalism of the text
Line 498 - S. miltiorrhiza - italic
Thank you for your contribution and the efforts you have put into this study.
Author Response
I am pleased to inform you that this is a well-executed manuscript. The paper is well-structured, and the topic addressed is relevant to the field.
Thank you for your recognition of our study. We have revised the paper carefully based on your comments. We hope that our revisions have addressed all the comments satisfactorily.
Comments 1: I noticed that most of the keywords are already included in the title. To enhance the discoverability of the manuscript and improve its indexing, I recommend revising the keywords to include terms or phrases that complement the title but are not explicitly mentioned in it.
Response 1: Thanks for the suggestion. We have added two keywords in lines 29-30.
Comments 2: Line 64 - Replace Salvia miltiorrhiza whit S. miltiorrhiza
Response 2: Thanks for the suggestion. We have replaced Salvia miltiorrhiza whit S. miltiorrhiza in line 59.
Comments 3: Line 164 – Replace Gossypium hirsutum with G. hirsutum
Response 3: Thanks for the suggestion. We have replaced Gossypium hirsutum with G. hirsutum in line 157.
Comments 4: In general, the full Latin name of each species be written out in its entirety the first time it appears in the text. Subsequent mentions can then use the abbreviated form consistently throughout the manuscript. You have largely followed this rule; however, I noticed some inconsistencies in a few instances. Please review the manuscript to ensure that this rule is applied consistently to all species, as this will enhance the uniformity and professionalism of the text
Response 4: Thanks for the suggestion. We have checked and revised the full text.
Comments 5: Line 498 - S. miltiorrhiza – italic
Response 5: Thanks for the suggestion. We have italicized “S. miltiorrhiza” in line 469.

This manuscript is a resubmission of an earlier submission. The following is a list of the peer review reports and author responses from that submission.
Round 1
Reviewer 1 Report
Comments and Suggestions for Authors
The KNOXs are conserved transcription factors in plants and known to play important roles in abiotic stresses. The authors isolated KNOX genes from S. miltiorrhiza, an herbaceous plant used as a traditional Chinese medicine, and compared them with other plants’ KNOXs. The authors also examined the transcriptional responses of these KNOXs to salt stress and found that the expression of SmKNOX4 was decreased while SmKNOX1, 2, 8, and 9 increased within 24 h after salt exposure.
The analyses seem to be performed proper ways and the descriptions look fine overall. However, the presented data do not directly reveal the functional role(s) of SmKNOXs in the stress responses of S. miltiorrhiza. For example, the authors may have been able to show the possible target genes of SmKNOXs and examine their expression patterns compared to SmKNOXs. Such information would emphasize the value of this manuscript.
Also, authors need to provide more detailed method of salt stress treatment. For example, the age of plant, growth condition (temperature, day length, light strength, whether the plants were grown in the medium or in soil?, etc). Photographs (or illustrations) of plants will be helpful for readers’ understanding.
Reviewer 2 Report
Comments and Suggestions for Authors
In this article, Deng and his (her) colleagues conducted a genome-wide identification of all KNOX genes in Salvia miltiorrhiza and examined the expression profiles of KNOX genes in different tis sues and under salt stresses to explore their function. The manuscript enriches our knowledge about the role of KNOX in S. miltiorrhiza. The conclusions are supported by the data, and the submitted manuscript is general interest to the readers. However, I have several comments that should be addressed before publication.
In scientific aspects:
1. In the section 2.5 (Cis-acting element analysis), cis-acting elements are lying the promoters of the genes, So the author should give an accurate description instead of confused description (such as 2000 bp sequence upstream of the SmKNOX genes, from ATG or other part of the gene?), please check and correct them to improve the manuscript.
2. In the section 2.7 (Expression analysis of SmKNOXs under Salt Stress), the authors should introduce the tissue of the gene expression detected, or introduced in the 4.5 section (Materials and Methods part). The whole seedlings? Or the roots (significant medicinal value tissue)? Or the flowers? Or the stem?
Comments on the Quality of English LanguageIn language aspect:
1. I suggest that the “KNOX genes play an important role ……”(page1, L36) replaced by “KNOX gene play an important role ……” or “KNOX genes play important roles ……” . There are only the examples, please go through the manuscript and correct them.
2. I suggest that the “while were almost not expressed……” (page6, L187) replaced by “while they were almost not expressed……” or “but were almost not expressed……” .
3. I suggest that the “SmKNOX6 and 7……” (page9, L284) replaced by “SmKNOX6 and SmKNOX6……” . Similar expression should be corrected in the revised manuscript.
4. The authors must carefully check grammar, punctuation, spelling, and overall style in the whole text.
Reviewer 3 Report
Comments and Suggestions for Authors
Salvia miltiorrhiza is a herbaceous plant, and faces a variety of abiotic stresses, such as salt and drought during the cultivation process. In this study, the authors conducted a genome-wide identification and analysis of all KNOX genes in S. miltiorrhiza, and identified a set of KNOX genes might involved in the regulation in salt stress responses. The manuscript presents a linear and logical argument, there are several issues that should be addressed.
1. Genes should be italicized. Please check the full text.
2. In Figure 2, the top half of the picture is missing, please fill in the blanks.
3. Table 1 needs to be reformatted, with minimal pagination.
4. In Figure 6, KNOX3 and 5 are both upregulated, and these two genes also need to be validated.
5. Tissue expression profiles of genes need to be re-validated with Qrt-PCR.
6. To validate KNOX as a transcription factor, subcellular localisation experiments are required.
7. The conclusions need to be further refined as to which KNOX may play a role in salinity resistance in Salvia miltiorrhiza.
8. The comma in the title can be removed.
9. In section of Materials and methods, Gene expression analysis section lacks a detailed description of the sampling method, which will affect the reader's understanding of the results.
10. Authors need to standardize the format of references.
Comments on the Quality of English LanguageThe manuscript is challenging to read in numerous instances. It is recommended that the authors submit it to a native speaker for proofreading.